# Hearing Benefits of Clinical Management for Meniere’s Disease

**DOI:** 10.3390/jcm11113131

**Published:** 2022-05-31

**Authors:** Yi Zhang, Chenyi Wei, Zhengtao Sun, Yue Wu, Zhengli Chen, Bo Liu

**Affiliations:** 1Department of Otolaryngology Head and Neck Surgery, Beijing Tongren Hospital, Capital Medical University, Beijing 100730, China; zhangyi_trhos@mail.ccmu.edu.cn (Y.Z.); weichenyi_ent@163.com (C.W.); zhengtaosun@mail.ccmu.edu.cn (Z.S.); wuyue19940507@sina.com (Y.W.); chenzhengli@mail.ccmu.edu.cn (Z.C.); 2Beijing Institute of Otolaryngology, Key Laboratory of Otolaryngology Head and Neck Surgery, Ministry of Education, Beijing 100730, China

**Keywords:** Meniere’s disease, sensorineural hearing loss, clinical management, pure-tone audiometry

## Abstract

Meniere’s disease is a progressive hearing–disabling condition. Patients can benefit from strict clinical management, including lifestyle and dietary counseling, and medical treatment. A prospective cohort study was carried out with 154 patients with definite Meniere’s disease, with an average age of 43.53 ± 11.40, and a male to female ratio of 0.97:1. The pure-tone thresholds of all 165 affected ears, over a one-year clinical management period, were analyzed. After one year, 87.27% of patients had improved or preserved their hearing at a low frequency, and 71.51% at a high frequency. The hearing threshold at frequencies from 250 Hz to 2000 Hz had improved significantly (*p* < 0.001, *p* < 0.001, *p* < 0.001, *p* < 0.01), and deteriorated slightly at 8000 Hz (*p* < 0.05). Of all the patients, 40.00% had a hearing average threshold that reached ≤25 dB HL after the clinical management period, among whom 27.27% were patients in stage 3. The restoration time was 2.5 (1.0, 4.125) months, with a range of 0.5–11.0 months, and the restoration time was longer for stage 3 than for stages 1 and 2 (*u* = −2.542, *p* < 0.05). The rising curves improved the most (*p* < 0.05), with most becoming peaks, whereas most peaks and flats remained the same. Patients who were initially in the earlier stages (95% CI 1.710~4.717, OR 2.840, *p* < 0.001), have an increased odds ratio of hearing by an average of ≤25 dB HL. Age (95% CI 1.003~1.074, OR 1.038, *p* = 0.031), peak curve (95% CI 1.038~5.945, OR = 2.484, *p* = 0.041), and flat curve (95% CI 1.056~19.590, OR = 4.549, *p* = 0.042), compared with the rising curve, increase the odds ratio of hearing on average by >25 dB HL. Most patients can have their hearing preserved or improved through strict clinical management, and sufficient follow-up is also essential. Stage 3 patients also have the potential for hearing improvement, although the restoration time is longer than in the early stages. The initial hearing stage, age, and audiogram pattern are related to the hearing benefits.

## 1. Introduction

Meniere’s disease (MD) is a chronic unique inner ear disease associated with endolymphatic hydrops by histopathology, and is characterized by recurrent spontaneous vertigo, fluctuating sensorineural hearing loss (SNHL), tinnitus, and aural fullness [1]. Though not curable, patients with Meniere’s disease can experience symptom relief and prevent permanent damage to their hearing through clinical management.

The Bárány society defines Meniere’s disease as episodic vestibular syndrome in the International Classification of Vestibular Disorders (ICVD) [2]. The lesions of the auditory and vestibular system occur either simultaneously, or some precede others [3]. The effects of vertigo are always taken seriously; however, part of vertigo is the fluctuating and gradual decline in the hearing, which can potentially develop into a hearing disability. Hearing loss was included in the original description of this disease and is still considered a necessary criterion by the current international consensus. It is generally known that the audiogram has characteristic features and specific fluctuation patterns [4]. The hearing loss usually occurs at low frequencies and is accompanied by hearing fluctuations, which are characteristic of the early stages of Meniere’s disease [5,6,7]. In this period, most patients’ hearing loss is reversible; therefore, as the disease progresses, hearing tends to present full-frequency, moderate-to-severe hearing loss, and loss becomes permanent [8]. Those who have poor hearing often give up hopes of recovery. Nevertheless, residual hearing is vital. Progressive and occult hearing loss will often confuse patients and be ignored by doctors who have no complete knowledge system.

The treatment for Meniere’s disease consists of reducing attacks, relieving symptoms, and preventing permanent damage to hearing. Lifestyle changes, diuretics, vasodilators, corticosteroids, intratympanic therapy, and surgical treatment methods are preferred. Although there is currently no complete cure for Meniere’s disease, clinical management, which includes lifestyle and dietary changes, medical treatment, psychological counseling, or minimally invasive procedures, such as intratympanic steroid therapy, helps most patients [4,9,10]. Clinically, fluctuating and progressive sensorineural hearing loss is relatively refractory to treatment, whereas episodic vertigo is usually quite responsive [11]; however, the prevention of hearing disabilities is as important as the control of vertigo. The World Report on Hearing 2021 [12] revealed that nearly 1.5 billion people worldwide live with some degree of hearing loss, and nearly 430 million have moderate to severe hearing loss. Many studies have shown that more than half of Meniere’s patients had hearing loss at an advanced stage, as they are either at stage 3 or 4 [13,14]. Control of the hearing lesion is the main aim of the management of Meniere’s disease due to its great importance for the patients’ quality of life, whether for symptom control, hearing disability prevention, functional rehabilitation, and even a reduction in the economic burden on patients and society.

The repeated attack, fluctuation, and progression of Meniere’s disease are consistent with the modality of chronic disease; therefore, it also needs to be managed according to the clinical practice of chronic diseases. The importance of lifestyle adjustment is put forward in the guidelines for MD. We conducted a study with a small series of patients and found that the hearing restoration of MD patients takes months; however, that observation period was only three months. A more extended observation was needed. During this observation period, strict clinical management and close follow-ups were needed for practice. In this study, the term “clinical management”, in its broadest sense, refers to the management of the core components of lifestyle adjustment and medical treatment.

Moreover, hearing features have rarely been studied and reported in recent years after the treatment and criteria were updated; there is no clear consensus among different studies regarding the characteristics and evolution of audiometric thresholds and the morphology of the audiogram of patients at different stages of Meniere’s disease. In 1995, the American Academy of Otolaryngology-Head and Neck Surgery (AAO-HNS) devised a staging system based on the average hearing threshold of pure-tone audiometry (PTA) [15]. This staging system provides a relatively objective basis for the clinical classification of severity and emphasizes the importance of understanding the progression of hearing loss in this disease. According to previous studies, four main audiometric patterns have been described in patients with Meniere’s disease: (1) the ascending type with low-frequency decline, (2) the descending type with high-frequency decline, (3) the peak type with low-frequency and high-frequency decline, and (4) the flat type with total frequency decline [16]. An audiogram and its changing regularity [17,18,19] provide the recognizable audiological features of the disease. Our study is based on the above practical staging system and audiometric patterns; therefore, we followed up with a group of patients with Meniere’s disease, we analyzed and reported their hearing benefits after one year of strict clinical management, and investigated any hearing-related information that may be crucial for determining hearing prognosis in patients with Meniere’s disease.

## 2. Materials and Methods

### 2.1. Study Participants

This is a prospective cohort study with pre- and post- intervention comparisons with the same group of patients. These patients were diagnosed with definite Meniere’s disease for whom audiological data were collected in the Hearing and Vestibular Clinic of the Department of Otorhinolaryngology of the tertiary hospital from August 2017 to November 2020. All procedures carried out in studies involving human participants were consistent with the ethical standards of the Ethics Committee of the Hospital.

Inclusion criteria: (1) all patients were diagnosed with definite Meniere’s disease according to the diagnostic criteria approved by the Classification Committee of the Bárány Society (2015) [1]: (a) two or more spontaneous episodes of vertigo, each lasting 20 min to 12 h; (b) audiometrically documented low-to-medium frequency sensorineural hearing loss in one ear, defining the affected ear on at least one occasion before, during, or after one of the episodes of vertigo; (c) fluctuating aural symptoms in the affected ear; and (d) symptoms not better accounted for by another vestibular diagnosis. (2) Subjects also met the following hearing loss criteria: low-frequency hearing of the affected ear is defined as greater increases in pure-tone thresholds (i.e., worse) than the contralateral ear by at least 30 dB HL at each of the two contiguous frequencies below 2000 Hz. In cases of bilateral hearing loss, the absolute low-frequency thresholds must be 35 dB HL or higher at each of the two contiguous frequencies below 2000 Hz. (3) All the patients consented to receive continuous follow-ups for one year or longer.

Exclusion criteria: (1) the specified conditions to be excluded include, including vestibular migraines, superior semicircular canal dehiscence syndrome, sudden hearing loss, labyrinthitis, drug-induced deafness, posterior circulation ischemia, schwannoma, definite autoimmune inner ear disease, delayed endolymphatic hydrops, and intracranial tumor. (2) Patients with Meniere’s disease who had received an intratympanic gentamicin injection, endolymphatic sac surgery, semicircular canal occlusion, vestibular neurectomy, or cochlear implant surgery were excluded.

Ultimately, a total of 156 patients were enrolled in the study, for two of whom follow-up data were lost for not coming back on time. Finally, the hearing of 165 ears in 154 patients was analyzed. There was unilateral Meniere’s disease in 143 ears and bilateral Meniere’s disease in 22 ears, with an average age of 43.53 ± 11.40 (19–70 years old), consisting of 76 males (49.35%) and 78 females (50.65%), with a male-to-female ratio of 0.97:1 (Table 1).

### 2.2. Procedures

All the patients registered clinical information in detail, including sex and age. Medical histories were recorded in detail. Examination of otology was routinely performed. Pure tone audiometry was taken routinely. MRI of the internal auditory canal and posterior fossa was taken to exclude other diseases which can present vertigo and audiometrically verified asymmetric sensorineural hearing loss. Caloric and head impulse tests were used for the evaluation of the vestibular system’s status. The audiograms were analyzed as evidence.

Regarding clinical management, when they received the diagnosis, all patients received education regarding a low-sodium diet, avoidance of particular foods and products, such as alcohol, caffeine, monosodium glutamate, excessive tea, and so on. Healthy lifestyle modifications and psychological counseling are needed, such as quitting smoking, avoiding allergy triggers, getting plenty of sleep, maintaining a good mood, taking part in appropriate physical activities, and so on. Patients were encouraged to make notes relating to their health. During the treatment course, patients were provided pharmacologic treatment with necessary steroids, betahistine, diuretics, or ginkgo preparation for the days or months of the trial, in order to control symptoms and maintain a stable state. When patients experienced an acute attack, systemic steroid therapy or local steroid injection were options for treating patients’ hearing loss. The systemic steroid was given prednisone 5 mg/kg/d as an initial dosage, for approximately 10–14 days with a slow tapering dose, and a local steroid injection was performed instead if there were contraindications with the systemic use of the steroid or when used as a supplementary treatment. Moreover, the patients temporarily used vestibular inhibitors during acute vertigo attacks and used betahistine with other medications for vertigo control at other times. We used betahistine 36 mg/d, for the attack and interval therapy. The treatment duration depends on patients’ symptoms, including vertigo and hearing loss. Hence, some patients used betahistine for several months, usually 3 to 6. Hydrochlorothiazide 25 mg/d as diuretics were used when patients were refractory to steroids and betahistine. Ginkgo biloba preparations at 57.6 mg/d, were not opposed to if needed. Vestibular rehabilitation exercise was performed during the non-acute period if needed. Patients were encouraged to keep a health diary. In addition, a one-year clinical follow-up was conducted.

### 2.3. Audiometric Measurement

The pure-tone audiometric data were collected by professionals with a GSI 61 audiometer in a standard soundproof room, which limited the background noise to below 18 dB A. The audiometric thresholds for 250, 500, 1000, 2000, 4000, and 8000 Hz were determined by air and bone conduction, and narrow-band noise was used for masking.

The poorest audiogram within six months of the interval before entering the study determined the patients’ onset hearing level, and their poorest audiogram at the one-year point (12 months ± 0.5 months) was taken as the hearing level after the one-year follow-up. For the criterion of clinical hearing improvement or deterioration, a greater than 5 dB change was used as an outcome measure [5,20]. The pure-tone threshold results were analyzed using two groups of frequencies: (1) low frequency (the average threshold value of 250, 500, and 1000 Hz) and (2) high frequency (the average threshold value of 4000 and 8000 Hz). As the Bárány Society (2015) [1] and the AAO-HNS Committee on Hearing and Equilibrium criteria for Meniere’s disease (1995) [15] have recommended, we finally staged the hearing from stage 1 to stage 4 using the average threshold corresponding to ≤25 dB HL, 26–40 dB HL, 41–70 dB HL, and >70 dB HL, respectively. In this study, staging was applied to evaluate the status of patients before treatment. During the one year of follow-up, the patients’ hearing was re-evaluated in order to draw comparisons.

To classify different types of the audiometric curve objectively, we referred to the function that Mateijsen and his colleagues [16] used to classify the curves, and made fine-tuning corrections according to the methods of other studies found in the literature [21,22]. Two professionals performed the fine-tuning. If the advice reached an agreement, the correction was made; however, if there was no consensus, the final result was based on the formula.

### 2.4. Statistical Analysis

The database was established using Excel 2016, and statistical analysis was conducted using IBM SPSS Statistics for Windows, Version 24.0 (IBM Corp., Armonk, NY, USA). MATLAB was used for visualization. The data presented a normal distribution using the Kolmogorov–Smirnov test and are expressed as the mean and standard deviation; Student’s *t*-test and a chi-square test were used for comparisons between groups. Nonnormally distributed data were represented by quartile spacing and the Wilcoxon signed-rank test, whereas Spearman’s rank–order correlation test was used for comparisons between groups. Binary logistic regression was used to analyze risk factors. *p* < 0.05 was considered statistically significant.

## 3. Results

### 3.1. Initial Hearing

Affected ears at stage 3, and with a peak curve, represented the greatest proportion of all the collected hearing data. Table 2 showed the distribution of different stages and audiometric curves of these patients. The proportion of patients at stage 3 reached their highest at 49% (80/165); the rest were 17% (28/165) at stage 1, 32% (53/165) at stage 2, and 2% (4/165) at stage 4, respectively. The peak curves accounted for the greatest proportion in each stage. Moreover, the proportion of flat curves rose from stage 1 to stage 4. There was no difference in the composition of the curve type of each stage (*χ^2^* = 14.064, *p* = 0.08). There was a significant correlation between the average threshold of low and high frequencies (*r* = 0.433, *p* < 0.001), and the coefficient of high frequency:low frequency was 0.676:1.

### 3.2. Hearing Benefits after Clinical Management

#### 3.2.1. Hearing Benefits and Hearing Restoration

Of the affected ears, 87.27% had improved or preserved hearing at a low frequency (66.06% improved and 21.21% preserved), and 71.51% at a high frequency (24.24% improved and 47.27% preserved). In Figure 1, a clear, steady hearing improvement can be seen at 250 Hz (*z* = −8.330, *p* < 0.001), 500 Hz (*z* = −7.391, *p* < 0.001), 1000 Hz (*z* = −5.856, *p* < 0.001), and 2000 Hz (*z* = −2.845, *p* = 0.004). Notably, at 4000 Hz, there is no significant difference (*z* = −0.263, *p* = 0.792). In contrast, the 8000 Hz threshold showed a slight deterioration (*z* = −2.001, *p* = 0.045).

The most striking result that emerged from the data is that 66 affected ears (40.00%) were restored to, or preserved at, a normal hearing level (average threshold ≤ 25 dB HL). The restoration time was 2.5 (1.0, 4.125) months, with a range of 0.5–11.0 months. The restoration time of the advanced stages (stages 3 and 4) (3.0 (1.75, 6.25 months)) was longer than that of the early stages (stages 1 and 2) (2.0 (0.56, 3.00 months)), and the differences were statistically significant (*u* = −2.542, *p* = 0.011). The patients with hearing thresholds that reached ≤25 dB HL were younger than those with a hearing average >25 dB HL. Whether the hearing reached ≤25 dB HL or not had no relationship to sex, but there was a significant difference with regard to the initial audiometric curve patterns (Table 3).

#### 3.2.2. Hearing Benefits for Different Stages

The advanced stages still have potential for improvement. Figure 2 reveals that there were steady hearing benefits in the most affected ears after the one-year clinical management period. At the low frequency, the hearing at stage 2 (t = 4.945, *p* < 0.001) and stage 3 (t = 8.349, *p* < 0.001) improved significantly, whereas at stage 1 (t = 1.166, *p* > 0.05) and stage 4 (t = 2.818, *p* = 0.067), it did not show a significant difference. At 2000 Hz, at stage 1 (t = −2.228, *p* = 0.034), hearing deteriorated, and at stage 3 (t = 4.518, *p* < 0.001), it improved, whereas the changes in the other two stages did not show a significant difference (*p* > 0.05). At the high frequency, at stage 1 (t = −2.273, *p* = 0.031), hearing deteriorated, and at stage 4 (t = 4.371, *p* = 0.022), it improved significantly, whereas the change in the other two stages did not show a significant difference (*p* > 0.05). In terms of average hearing threshold value, at stage 2 (t = 3.283, *p* = 0.002), hearing deteriorated slightly, and at stage 3 (t = 7.436, *p* < 0.001), it improved slightly. The hearing of patients at advanced stages (stages 3 and 4) improved more both at low (t = 3.409, *p* = 0.001) and high (t = 2.333, *p* = 0.021) frequencies than that of patients at an early stage (stages 1 and 2).

What stands out in Figure 3 is that the majority of the affected ears had an improved or preserved hearing level. For the individuals, the hearing of the majority (67.86%) at stage 1 was still ≤25 dB HL, whereas a small proportion had progressed to stage 2 (21.43%) or stage 3 (10.71%); nearly half (54.72%) of the affected ears at stage 2 improved their hearing to ≤25 dB HL, 26.42% remained at 26–40 dB HL, and 18.87% progressed to 41–70 dB HL; half (50.00%) of the stage 3 patients remained at 41–70 dB HL, 22.50% improved to <25 dB HL, 26.25% improved to 26–40 dB HL, and 1.25% progressed to >70 dB HL. Of the four affected ears at stage 4, two improved to 41–70 dB HL, and the other two remained at >70 dB HL.

#### 3.2.3. Hearing Benefits for Different Audiometric Curves

Figure 4 shows that there were steady hearing benefits in most patients with different audiometric curves. At the low frequency, all types improved significantly (i.e., from 50.49 dB HL to 30.63 dB HL (t = 9.688, *p* < 0.001), 42.41 dB HL to 35.29 dB HL (t = 2.390, *p* = 0.024), 44.79 dB HL to 35.49 dB HL (t = 4.555, *p* < 0.001), and 56.67 dB HL to 47.35 dB HL (t = 2.336, *p* = 0.033)) by means of rising, falling, peak, and flat curves, respectively. At the high frequency, there was a slight deterioration in the rising (t = −2.032, *p* = 0.048) and peak curves (t = −1.215, *p* > 0.05), and a slight improvement in the falling (t = 1.333, *p* = 0.193) and flat curves (t = 1.065, *p* = 0.303), but there was no significant difference between various types (F = 2.539, *p* = 0.058).

What can be clearly seen in Figure 5 is the general pattern of the changes in the audiometric curves. For 46.48% of individuals, the rising curves at onset were likely to turn to peak curves after one year; for 14.58%, rising curves were maintained as rising curves; 14.58% showed a change to falling curves; and 29.17% showed a change to flat curves. The majority (89.66%) of the falling curves maintained the same type of curve. Nearly half (46.48%) of the peak curves were maintained as peak curves, 30.99% became falling curves, 18.31% became flat curves, and 4.23% became rising curves. Nearly half (47.06%) of the flat curves remained flat, whereas 29.41% became falling curves, 17.65% became peak curves, and 5.88% became rising curves.

#### 3.2.4. Multiple Factors Analysis of Hearing Benefits Based on Pure-Tone Audiometry

The dependent variable was defined as whether the average thresholds reached ≤25 dB HL after clinical management, and the independent variables and covariates included sex (male, female), age, audiometric curve (four types), and the initial stage (stage 1, 2, 3, and 4). Patients at an earlier stage initially (95% CI 1.710~4.717, OR 2.840, *p* < 0.001) had an increased odds ratio of hearing, on average, ≤ 25 dB HL. Age (95% CI 1.003~1.074, OR 1.038, *p* = 0.031) increased the odds ratio of hearing to an average of > 25 dB HL. Compared with the rising curve, the peak curve (95% CI 1.038~5.945, OR = 2.484, *p* = 0.041) and flat curve (95% CI 1.056~19.590, OR = 4.549, *p* = 0.042) increased the odds ratio of hearing to an average threshold of >25 dB HL. There was no relationship between sex and hearing restoration.

## 4. Discussion

In this study, we proposed a practical clinical management plan. Clinical and laboratory studies have indicated that Meniere’s disease is the clinical expression of inner ear changes caused by the progressive dysfunction of the accumulated endolymph and can progress to hearing disability. Although initial treatment includes lifestyle changes and pharmacologic therapy, many individuals eventually progress to hearing loss, but few lose hearing entirely, even in the later stages [23]; therefore, it is necessary to preserve the patient’s hearing as early as possible. The treatment is complicated since hearing, vertigo control, tinnitus, and suppression are all intertwined, and thus, are tricky to treat separately. Both pharmacologic and surgical treatment may be effective for some patients and ineffective for others [24,25]; however, there is no current cure for Meniere’s disease, which is especially crucial for sensorineural hearing loss. For surgical therapy, patients and doctors sometimes find a clinical course difficult to decide on. This does not mean an opposition to surgery, but we need to determine when and how to choose. Preserving the structure appropriately is of the same importance. Nevertheless, most patients will benefit from criterial clinical management.

In this study, pure-tone audiometry is used for data collection and deep analysis. The treatment of hearing in Meniere’s disease consists of a series of practical methods, which needs efficient management. High-efficiency diagnosis and evaluation are the key points; however, high-efficiency diagnosis and management remain challenging for clinicians. During the past decade, magnetic resonance imaging, electrocochleography, and many other techniques have been developed for the inner ear, leading to advancements in diagnosing Meniere’s disease; however, we still require a non-invasive, cost-effective, and universally available test with high sensitivity and specificity for Meniere’s disease. Pure-tone audiometry meets these requirements. The hearing loss pattern can provide information for future efforts to slow down its progress [26].

In our series of patients with definite Meniere’s disease, nearly half of the patients had moderate to severe hearing loss, and their average hearing thresholds displayed a predominance of stage 3 and stage 4. This observation was similar to that of others, which is noted in the literature [8]. The proportion of patients with moderate to severe hearing loss in this study was less than that previously published [14]. This may be due partly to the increased attention and improvement of the early diagnosis of Meniere’s disease. Furthermore, a focus on patient education could enable more patients to be aware of their symptoms and to see a doctor at an earlier stage.

Encouragingly, after one year of strict clinical management, most patients improved their hearing at a low frequency and/or maintained their hearing at a high frequency. Only a few had their hearing deteriorate at low and high frequencies. Thus, most patients benefitted from criterial clinical management and sufficient observation. This result is better than that of a former study, which suggested that the percentage of patients who experienced fluctuations resulting in hearing deterioration ranged from 58% to 67% [5]. Additionally, a slightly higher value than Molnár A’s result of intratympanic steroid therapy of 68.6% [27] was achieved. The different length of the follow-up period and the development of the criteria of the disease are possible reasons for the different results.

What is surprising, is that 40% of patients had their hearing thresholds reach ≤25 dB HL, even the 18 patients in stage 3. A long period, comprising months, was required for patients to restore their hearing. Their hearing did not always improve rapidly after the treatment, but it was gradually restored after months of waiting. This has not been reported before, and it subverts our inherent understanding of this issue; however, it provides essential data on the timing of hearing restoration. The restoration time of the advanced stage was longer than that of the earlier stages. The longest time was 11 months, the middle time for advanced stages was 3 months, and for the early stages, it was 2 months. According to this result, we recommend a hearing observation period for patients of Meniere’s disease of at least one year. Furthermore, compared with early stages (stages 1 and 2), patients’ hearing in advanced stages (stages 3 and 4) improved more at low and high frequencies. Patients in advanced stages should not be neglected, nor should they give up on rescuing their hearing. Instead, we emphasize that more attention should be paid to their symptoms, and they should attend follow-up appointments with their clinician to maintain their residual hearing. The hearing changes associated with Meniere’s disease need further observation to determine when to carry out an artificial hearing intervention or surgery, as irreversible hearing loss may occur.

The importance of examining individual frequency thresholds is emphasized as one evaluates and monitors patients with Meniere’s disease over time [28]. The proportion of patients with low-frequency hearing improvements was the highest. Low-frequency hearing in stages 1 to 4 improved, and the improvements in stages 2 and 3 were significantly different; however, few studies have focused on high-frequency hearing loss. Some studies [17] have suggested that high-frequency hearing is related to presbycusis and that the effect of age is greater than the effect of the duration of the disease, which means that an audiogram shows an ascending curve when age correction is carried out; however, we report here that the decline in the high-frequency hearing was not due to age. We found a correlation between the average threshold of low- and high-frequency hearing. As the low frequencies declined, the high frequencies also declined relatively slowly. After one year, the high frequency deteriorated at stages 1 and 2, especially in stage 1, with a significant difference; it improved at stages 3 and 4, especially at stage 4. This outcome indicates that the decline in high frequency is somewhat associated with changes in the pathological progress of the disease itself, instead of aging. This argument is similar to that of Albre [29] and colleagues, who concluded that age does not influence the course of hearing loss of a pure-tone average threshold in patients with Meniere’s disease.

The proportion of the peak curve was the greatest, which is similar to the findings of Paparella [18] and Lee [19], who reported a significant prevalence of the peak curve of 42.9% and 50.26%, respectively. The ascending-type curve is generally considered to be characteristic of Meniere’s disease; however, it is usually not the most common type of curve in practice. Some studies observed a predominantly flat audiometric curve [13,22,30], even in very early times [22]. Enander [22] and colleagues observed that flat curves accounted for 60% of all patients. Thomas [30] and colleagues found that flat curves accounted for 44.5–45.9%. Other studies observed that the peak curve is the most common, represented by hearing loss at low and high frequencies and an audiogram peak at approximately 2 kHz [18,19]. There is no consensus regarding the standard of curve type in previous studies. In our research, we referred to the classification and calculation method in the literature when judging the curve type to make the decision objectively. Moreover, we manually reviewed and corrected the calculation results, which made the judgment of the audiogram curve more unified and standardized to reduce bias. This is the virtue of this essay when studying the audiometric curve. All four types improved significantly for different curves, utilizing rising, falling, peak, and flat types for the low frequency. Although there was no significant difference for the high frequency, it still surprisingly showed a slight deterioration in the rising and peak curves, and showed a slight improvement in the falling and flat curves. Patients who had rising curves at onset were most likely to have a peak after one year. This may have contributed to the percentage of the rising curve being the highest, and the decrease in the hearing of high-frequency patients at stage 1. Most patients with the falling curve still had this type (89.66%). Approximately half of the patients with a peak curve maintained a peak curve (46.48%). Most patients with flat curves remained flat (47.06%). Savastano [21] and colleagues reported that with disease development, an audiometric pattern tends to develop from the peak shape to the flat shape in a long-term follow-up (i.e., for ten years). After 15 years of follow-ups, the rate of the appearance of the flat type of audiogram increased from 21% to 75% [13].

We speculate that hearing loss appears in the low frequency initially, then in the high frequency, and then in the entire frequency, gradually deteriorating in this way. New studies report that the hearing loss profile does not only affect the low frequency in the early stages; however, a flat sensorineural hearing loss was observed in patients with familial Meniere’s disease and mutations in OTOG and MYO7A [31]. In our group only three patients exhibited a flat pattern in stages 1 and 2, potentially related to the fact that there were only two patients with familial histories of Meniere’s disease. More specifically, although our results show that patients had satisfactory hearing improvement, in the long term, the hearing improvement might not necessarily be the result of any treatment but rather the result of fluctuation in Meniere’s disease itself. The allelic variant may associate with the progression of hearing loss in patients with MD, such as major histocompatibility complex class I chain-related A (MICA) [32] and Toll Like Receptor 10 (TLR10) [33]. The beginning of high-frequency hearing loss often occurs in the early stage, which is also part of the hearing fluctuation and the process of Meniere’s disease. Changes in different frequencies may provide suggestive information regarding the prognosis. High-frequency hearing loss in the first audiogram, age of onset, and migraine can help to assess the risk of bilateral SNHL in MD [34]. In our group, there were a few bilateral MD cases with no familial histories, and with an increasing number of patients, further investigations will be conducted. The improvement of sensorineural hearing loss is strongly associated with the prognosis of tinnitus [35], which argues for future research on aural symptoms in patients. The characterization of the manifestation of hearing loss is essential for appropriate counseling and an improved understanding of the disease. Sufficient longitudinal follow-up and hearing evaluation are cardinal when making clinical practice decisions.

More long-term follow-up studies are needed to investigate the progressive patterns of different patients and different treatment outcomes with Meniere’s disease. The hearing of the patients fluctuates alternatively between improvement and deterioration. The results of a five- or ten-year follow-up are expected in the future for more clinical information. Although vestibular migraines are excluded from the study, migraine symptoms are usually complex, and the influencing factors should be strictly limited when observing Meniere’s disease hearing changes. In the future, research needs to include more details. The influencing factors should be more strictly limited. Further in-depth comparative analysis should be conducted when the case numbers increase. Future research directions may also be highlighted in terms of predictive factors of listening outcomes.

## 5. Conclusions

Most patients can have their hearing preserved or improved through strict clinical management, and sufficient follow-up time is also essential. Stage 3 patients also have potential for improvement, and their restoration time is longer than in the early stages. Initial hearing stage, age, and audiogram pattern are related to the hearing benefits.

## Figures and Tables

**Figure 1 jcm-11-03131-f001:**
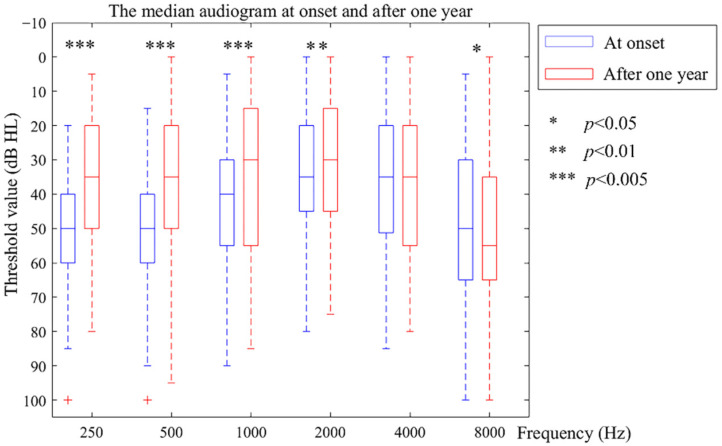
The medium audiogram at onset and after one year.

**Figure 2 jcm-11-03131-f002:**
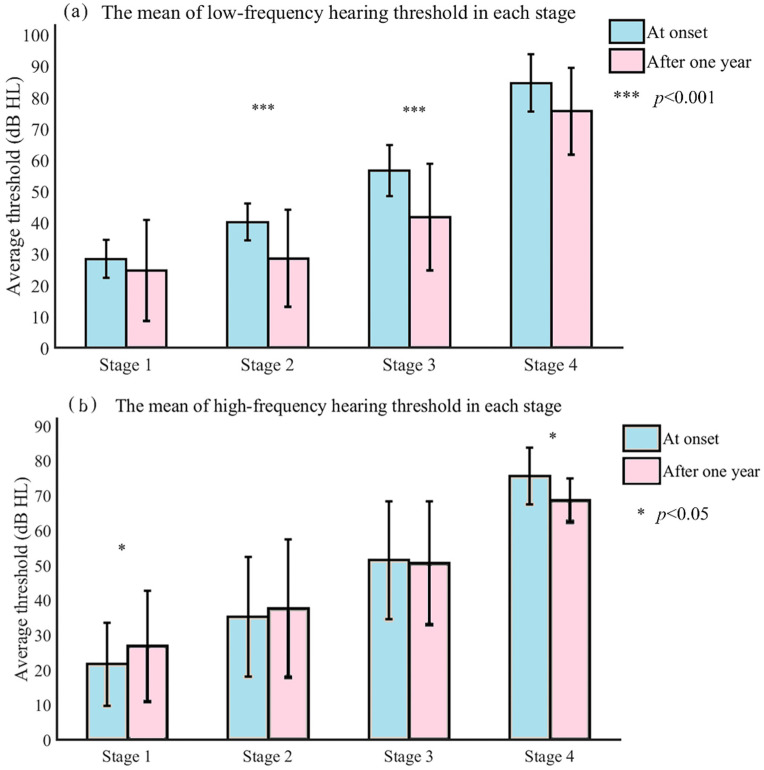
The mean low-frequency and high-frequency hearing thresholds at onset and after one year for each stage.

**Figure 3 jcm-11-03131-f003:**
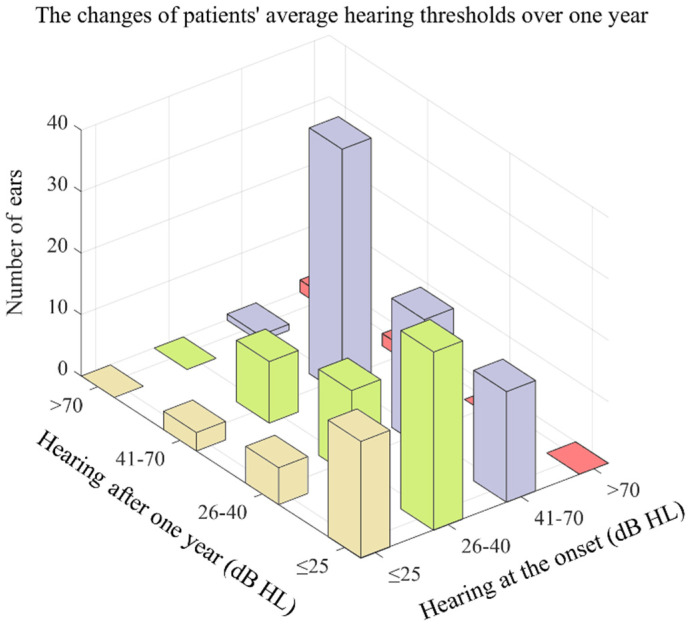
The changes in patients’ average hearing thresholds over one year.

**Figure 4 jcm-11-03131-f004:**
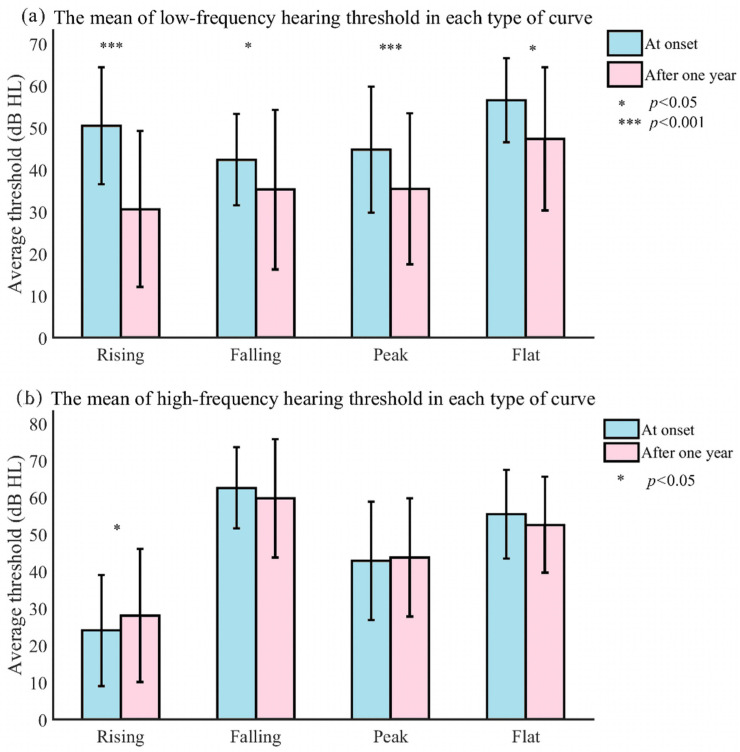
The mean of the low- and high-frequency hearing thresholds at onset and after one year of each type of curve.

**Figure 5 jcm-11-03131-f005:**
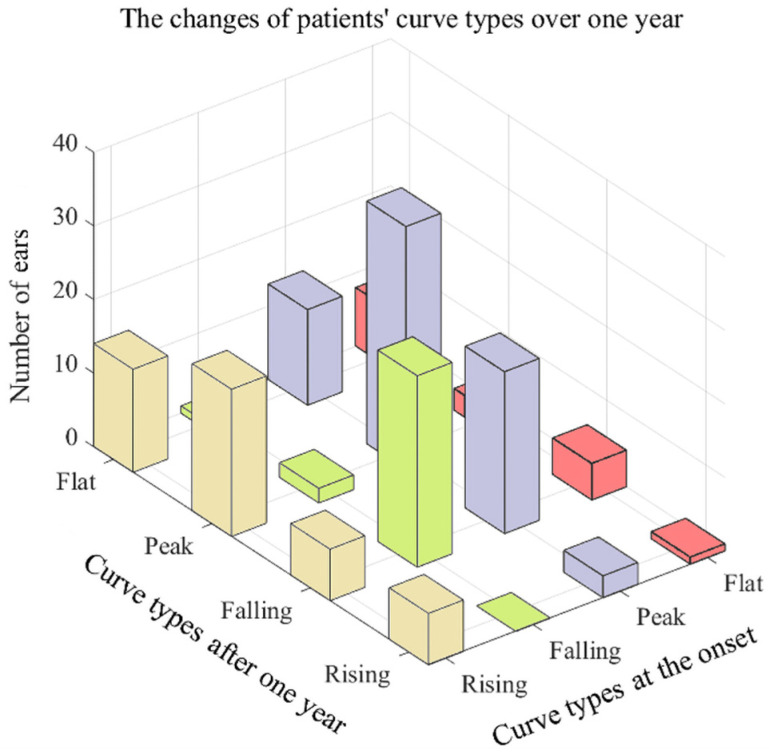
Changes in patients’ curve types over one year.

**Table 1 jcm-11-03131-t001:** Overview of the patients with definite Meniere’s disease.

Clinical Features	Unilateral MD	Bilateral MD	*p*-Value
Age (year) (mean ± SD)	43.24 ± 11.56	47.27 ± 8.60	0.260 ^•^
Sex (n)			
Male	70	6	0.764 ^†^
Female	73	5	
Sides (n)			
Right	67	11	0.822 ^†^
Left	76	11	
Comorbidities (n)			
Migraine	36	1	0.403 °
Familial Meniere’s history	2	0	1.000 *
Autoimmune disease	6	2	0.190 °
Duration of the disease (months) (Median, IQR)	16 (6, 27)	36 (6, 48)	0.293 ^#^
Initial hearing average (dB HL)	41.84 ± 15.76	40.15 ± 12.90	0.632 ^•^
Initial stage (n)			
1	24	4	1.000 *
2	46	7	
3	69	11	
4	4	0	

^•^ Student’s *t*-test; **^†^** Chi-square test; * Fisher’s test; ° Continuity Correction; ^#^ Mann–Whitney U test.

**Table 2 jcm-11-03131-t002:** The number and proportion of each stage and curve type (n (%)).

	Rising	Falling	Peak	Flat	Total
Stage 1	10 (36)	2 (7)	16 (57)	0 (0)	28 (100)
Stage 2	14 (26)	11 (21)	25 (47)	3 (6)	53 (100)
Stage 3	23 (29)	16 (20)	28 (35)	13 (16)	80 (100)
Stage 4	1 (25)	0 (0)	2 (50)	1 (25)	4 (100)
Total	48 (29)	29 (18)	71 (43)	17 (10)	165 (100)

**Table 3 jcm-11-03131-t003:** The comparison between different hearing results of the patients after clinical management.

	Group	Restored Average Thresholds ≤ 25 dB HL	Restored Average Thresholds > 25 dB HL	*p*-Value
N		66	99	
Age (year)		40.05 ± 10.70	46.26 ± 11.01	<0.001 ^•^
Sex (n)				
	male	30	52	0.428 ^†^
	female	36	47	
Initial stage (n)				
	1	19	9	<0.001 *
	2	29	24	
	3	18	62	
	4	0	4	
Initial curve (n)				
	rising	28	20	0.007 ^†^
	falling	8	21	
	peak	27	44	
	flat	3	14	

^•^ Student’s *t*-test; **^†^** Chi-square test; * Fisher’s test.

## Data Availability

The data presented in this study are available on request from the corresponding author. The data are not publicly available because the health examination data of a private hospital were used.

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
