# Peer review of "Hearing Benefits of Clinical Management for Meniere’s Disease"

_jcm, 2022, doi:10.3390/jcm11113131_

Round 1

Reviewer 1 Report

This prospective study compares audiograms in 165 Meniere's patients at presentation and after one year of standard clinical management.

There is no control group. The results could be solely attributed to the regression to the mean effect. The conclusions in the manuscript and in the abstract are therefore not backed by data.

Author Response

Dear Sir or Madam:

Thank you for taking the time to review the manuscript and thank you for your suggestion. We have carefully considered the suggestion and response to the comments as follows:

Point 1: There is no control group. The results could be solely attributed to the regression to the mean effect. The conclusions in the manuscript and in the abstract are therefore not backed by data.

Response 1:

I am sorry that this part was not clear in the original manuscript. I should explain it clearly. This study aims to observe and report the hearing outcomes of patients through strict clinical management. This kind of clinical management includes medicine implementation and health education and supervision(such as lifestyle and dietary changes, psychological counseling, etc.) for Meniere's disease patients. The hearing before and after the management were compared.

This is a self-controlled study, not a comparison of different treatment effects. When the patients get a definite diagnosis, they will receive help from doctors. Ethics opposes the lack of well-known treatment and essential healthy life guidance for the patients. And the management also fits the practice requirement described in the Clinical Practice Guideline: Meniere's Disease (2020, American). So, based on ethics, the control group was not set up in this study.

This paper is based on the data's mon factor and multiple factor analysis. The analysis includes both the paired and non-paired comparison and the individual frequency and average hearing analysis.

Considering the suggestion, we add some words to make it more clear in the method section: “This is a prospective self-control study of patients with a diagnosis of definite Meniere's disease……” (Please see in abstract line 13 and in the method section line 101)

We revised our English wording and had our manuscript polished by English-language editing service.

Thank you very much for taking the time to comment, promoting our expression more clearly and getting improvement.

Best Regard,

Yours sincerely

Reviewer 2 Report

This is a case series of 154 patients with Meniere disease in China.

The clinical description of the patients is incomplete since the authors do not report comorbidities such as migraine, autoimmune disease or history of familial Meniere disease. In addition, the authors must include if patients show bilateral involvement (bilateral hearing loss) and the duration of the disease. These data are essential to classify patients according to clinical criteria and compare the outcome and the authors should include them in a Table.

In the results there are no data on vestibular function. Please try to include them.

The discussion section is somehow incomplete. Hearing loss profile is not only affecting low frequency in the early stages. It may also show a flat sensorineural hearing loss in other subgroups of patients. This occurs in patients with familial Meniere disease and mutations in OTOG and MYO7A.

Several clinical factors and genetic biomarkers have been reported in hearing loss progression in Meniere disease such as MICA or TLR10, but this is missing in the discussion. I recommend the author to perform a new review of the published literature of hearing loss in Meniere disease and update the references.

A conclusion in the abstract is missing.

The conclusions should be presented as a numbered list

Suggested references

  1. Frejo L, Martin-Sanz E, Teggi R, et al. Extended phenotype and clinical subgroups in unilateral Meniere disease: A cross-sectional study with cluster analysis. Clin Otolaryngol. 2017;42(6):1172-1180. doi:10.1111/coa.12844
  2. Moleon MDC, Torres-Garcia L, Batuecas-Caletrio A, et al. A Predictive Model of Bilateral Sensorineural Hearing Loss in Meniere Disease Using Clinical Data. Ear Hear. 2021;Publish Ah:1-7. doi:10.1097/AUD.0000000000001169.
  3. Roman-Naranjo P, Gallego-Martinez A, Soto-Varela A, et al. Burden of Rare Variants in the OTOG Gene in Familial Meniere’s Disease. Ear Hear. 2020;41(6):1598-1605. doi:10.1097/AUD.0000000000000878.

Author Response

Dear Sir or Madam:

Thank you for taking the time to review the manuscript and thank you for the insightful suggestion. We have carefully considered the suggestion and response to the comments as follows:

Point 1: The clinical description of the patients is incomplete since the authors do not report comorbidities such as migraine, autoimmune disease or history of familial Meniere disease. In addition, the authors must include if patients show bilateral involvement (bilateral hearing loss) and the duration of the disease. These data are essential to classify patients according to clinical criteria and compare the outcome and the authors should include them in a Table.

Response 1:

Thank you for your rigorous suggestion. We add the comorbidities including migraine, autoimmune disease and history of familial Meniere disease in the table (Please see in Table 1). And we add the duration of the disease of Unilateral and Bilateral MD in the table (Please see in Table 1).

Point 2: In the results there are no data on vestibular function. Please try to include them.

Response 2:

Thank you for your suggestion, vestibular function is also important to the patient's evaluation. But in our opinion, in this paper, we focus on and emphasize the hearing of the patients, without discussing vertigo and vestibular function aspect. So, we propose not to enroll vestibular function in this paper. It will make the theme of the article more prominent.

Point 3: The discussion section is somehow incomplete. Hearing loss profile is not only affecting low frequency in the early stages. It may also show a flat sensorineural hearing loss in other subgroups of patients. This occurs in patients with familial Meniere disease and mutations in OTOG and MYO7A.

Several clinical factors and genetic biomarkers have been reported in hearing loss progression in Meniere disease such as MICA or TLR10, but this is missing in the discussion. I recommend the author to perform a new review of the published literature of hearing loss in Meniere disease and update the references.

 Response 3:

Thank you very much for pointing out the weakness in the discussion section. We add this important discussion in the paper. (Please see in the discussion section line 405-410 and 413-415 ).

Point 4: A conclusion in the abstract is missing.

Response 4:

We add the conclusion of “Initial hearing stage, age, and audiogram pattern relate to the hearing benefits.” (Please see in the abstract line 30 and the conclusion section lin 440).

Point 5: The conclusions should be presented as a numbered list

Response 5:

We revised the conclusions reference to the article format of JCM.

Point 6: Suggested references

Response 6:

We updated the references( Reference 31 to 34. Line521-532).

Thanks again for your rigorous advice, and we also hope that our edits and the responses satisfactorily address the issues and concerns you have noted.

Best Regard,

Yours sincerely

Reviewer 3 Report

Dear authors of "Hearing benefits of clinical management for Meniere's Disease",

While I appreciate the meticulous decision to showcase the hearing loss and its improvement with medical therapy, I do find that the study has a large gap in its methods. If the authors had included a control group, one without any medical mangement, then the hearing outcomes would be more clinically significant. Moreover, these audiogram patterns/types were never stratified to the treatment modalities. Without this information, the conclusions of this study are not applicable for the general population, given that it is not replicable in society without knowing which treatments each group had. 

There are also significant grammar issues throughout.

Thank you

Author Response

Dear Sir or Madam:

Thank you for taking the time to review the manuscript and thank you for your suggestion. We have carefully considered the suggestion and response to the comments as follows:

Point 1: While I appreciate the meticulous decision to showcase the hearing loss and its improvement with medical therapy, I do find that the study has a large gap in its methods. If the authors had included a control group, one without any medical mangement, then the hearing outcomes would be more clinically significant.

Response 1:

I am sorry that this part was not clear in the original manuscript. I should explain it clearly. This study aims to observe and report the hearing outcomes of patients through strict clinical management. This kind of clinical management includes medicine implementation and health education and supervision(such as lifestyle and dietary changes, psychological counseling, etc.) for Meniere's disease patients. The hearing before and after the management were compared.

This is a self-controlled study, not a comparison of different treatment effects. When the patients get a definite diagnosis, they will receive help from doctors. Ethics opposes the lack of well-known treatment and essential healthy life guidance for the patients. And the management also fits the practice requirement described in the Clinical Practice Guideline: Meniere's Disease (2020, American). So, based on ethics, the control group was not set up in this study.

Considering the suggestion, we add some words to make it more clear in the method section: “This is a prospective self-control study of patients with a diagnosis of definite Meniere's disease……” (Please see in the abstract line 13 and in the method section line 101)

Point 2:Moreover, these audiogram patterns/types were never stratified to the treatment modalities. Without this information, the conclusions of this study are not applicable for the general population, given that it is not replicable in society without knowing which treatments each group had.

Response 2:

These audiogram patterns/types are attributed to the endolymphatic hydrops of the disease and changes during the course, but not the different severity. So, the treatment provided does not depend on the audiogram patterns/types but on the acute and intermittent period and patients‘ response to treatment. All the patients received the clinical management including many aspects, but not only one. So the audiogram patterns/types were never stratified to the treatment modalities, and there is no treatment subgroup.

 Point 3: There are also significant grammar issues throughout.

 Response 3:

We have our manuscript underwent the English editing as suggested.

Thank you very much for taking the time to comment, promoting our expression more clearly and getting improvement.

Best Regard,

Yours sincerely

Round 2

Reviewer 2 Report

The authors have improved the submission and responded to all my questions.

Author Response

Thank you very much for the rigorous advice in making our paper developed.

Reviewer 3 Report

Dear Authors,

Thank you for your clarification. I see now that the is a self-controlling study. 

Few more edits:

1. Please describe what 'value' is in table 1 and 3. 

2. Also what are the ( ) noting in table 1?

3. English is still rough, like line 161, "got diagnosis" is awkward. "received diagnosis" is must better. There are numerous other grammatical errors. Including lines 474-477, very awkward to read. Please fix. 

4. Section 2.2: please provide the dosing and regimen of all the medications these patients took, including steroids and Ginkgo Biloba etc.

5. The discussion should mention the limitations. A severe limitation is that there are concurrent vestibular migraine diagnoses.

6. In the discussion, the future direction should include stratifying the audiograms based on the treatment regimens, so it isn't grouping all the ginko biloba and steroids together. 

Author Response

Dear Sir or Madam:

Thank you for taking the time to review the manuscript and thank you for your suggestion. We have carefully considered the suggestion and response to the comments as follows:

Point 1:  Please describe what 'value' is in table 1 and 3.  

Response 1:

Thank you for your suggestion. I have changed the p to p-Value in the table1 and 3.

There are symbol illustrations at the bottom of the table to explain what statistical method each value comes from. (please see the bottom of table 1 and 3)

For table 1. •Student’s t-test; †Chi-square test; * Fisher’s testï¼›°Continuity Correction; #Mann–Whitney U test.

For table 3. •Student’s t-test; †Chi-square test; * Fisher’s test.

Point 2:   Also what are the ( ) noting in table 1?

Response 2:

Thank you for pointing out some unclear information. I have revised them in the table.(please see table 1)

Age (year)(mean±SD)

Duration of the disease (months) (Median, Quartile)

Point 3: English is still rough, like line 161, "got diagnosis" is awkward. "received diagnosis" is must better. There are numerous other grammatical errors. Including lines 474-477, very awkward to read. Please fix.

Response 3:

I revised some words and sentences. Please check the line in the revised version line 144.434-436

"got diagnosis" was revised to "received the diagnosis" (please see in the revised version line 144.)

“Although vestibular migraine is excluded in the study, migraine symptoms are very complex, and the influencing factors should be strictly limited when observing Meniere's disease hearing changes. In the future, research needs to include more details. The influencing factors should be more strictly limited. Further in-depth comparative analysis should be conducted when the case numbers increase. Future research directions may also be highlighted in terms of predictive factors of listening outcomes.” (please see in revised version line 434-442)

Point 4: Section 2.2: please provide the dosing and regimen of all the medications these patients took, including steroids and Ginkgo Biloba etc.

Response 4:

The procedure part was revised as follows:

“The systemic steroid was given prednisone 5mg/kg/d as initial dosage, for approximately 10-14 days with a slow tapering dose. We used betahistine 36mg/d, for the attack and interval therapy. ……The treatment duration depends on patients' symptoms, including vertigo and hearing loss. Hence, some patients used betahistine for several months, usually 3 to 6. Hydrochlorothiazide 25mg/d as diuretics were used when patients were refractory to steroid and betahistine. Ginkgo biloba preparations 57.6mg/d were not opposed if needed.”

(please see revised version line.154-163)

Point 5:The discussion should mention the limitations. A severe limitation is that there are concurrent vestibular migraine diagnoses.

Response 5:

I have added the limitations in the discussion part as follows:

“Although vestibular migraine is excluded in the study, migraine symptoms are very complex, and the influencing factors should be strictly limited when observing Meniere's disease hearing changes.” (please see in revised version line 433-435)

Point 6: In the discussion, the future direction should include stratifying the audiograms based on the treatment regimens, so it isn't grouping all the ginko biloba and steroids together.

Response 6:

Thank you for your suggestion. I rewrote some sentences in the discussion part as follows:

“Although vestibular migraine is excluded in the study, migraine symptoms are very complex, and the influencing factors should be strictly limited when observing Meniere's disease hearing changes. In the future, research needs to include more details. The influencing factors should be more strictly limited. Further in-depth comparative analysis should be conducted when the case numbers increase.” (please see in revised version line 433-440)

Thank you for your valuable suggestion to make our paper more rigorous and get improved.

Best regards,

Yours sincerely